# Effects of Primary Aldosteronism and Different Therapeutic Modalities on Glucose Metabolism

**DOI:** 10.3390/jcm8122194

**Published:** 2019-12-12

**Authors:** Mi Kyung Kwak, Jee Yang Lee, Beom-Jun Kim, Seung Hun Lee, Jung-Min Koh

**Affiliations:** 1Division of Endocrinology and Metabolism, Hallym University Dongtan Sacred Heart Hospital, Dongtan, 7, Keunjaebong-gil, Hwaseong 18450, Gyeonggi-do, Korea; creamtea38@gmail.com; 2Division of Endocrinology and Metabolism, Asan Medical Center, University of Ulsan College of Medicine, 88, Olympic-ro 43-gil, Songpa-gu, Seoul 05505, Korea; asanmedicalamc@gmail.com (J.Y.L.); umkbj0825@amc.seoul.kr (B.-J.K.)

**Keywords:** aldosterone, primary aldosteronism, diabetes mellitus, pre-diabetes, adrenalectomy

## Abstract

Despite findings that aldosterone impairs glucose metabolism, studies concerning the effect of primary aldosteronism (PA) and its treatment on glucose metabolism are controversial. We aimed to determine glucose metabolism in PA and the effect of the treatment modality. We compared glucose metabolism between PA patients (*N* = 286) and age-, sex-, and body mass index-matched controls (*N* = 816), and the changes in glucose metabolism depending on the treatment modality (adrenalectomy vs. spironolactone treatment). Hyperglycemia including diabetes mellitus (DM; 19.6% vs. 13.1%, *p* = 0.011) was more frequent in PA patients. Hyperglycemia was also more frequent in PA patients without subclinical hypercortisolism (SH: *p* < 0.001) and in those regardless of hypokalemia (*p* < 0.001–0.001). PA patients and PA patients without SH had higher DM risk (odds ratio (OR); 95% confidence interval (CI): 1.63; 1.11–2.39 and 1.65; 1.08–2.51, respectively) after adjusting confounders. In PA patients, there was significant decrease in the DM prevalence (21.3% to 16.7%, *p* = 0.004) and fasting plasma glucose (*p* = 0.006) after adrenalectomy. However, there was no significant change in them after spironolactone treatment. Adrenalectomy was associated with more improved glucose status than spironolactone treatment (OR; 95% CI: 2.07; 1.10–3.90). Glucose metabolism was impaired in PA, regardless of hypokalemia and SH status, and was improved by adrenalectomy, but not spironolactone treatment.

## 1. Introduction

Primary aldosteronism (PA) is characterized by inappropriately high aldosterone production for the sodium status, due to an aldosterone-producing adenoma (APA), unilateral adrenal hyperplasia (UAH), or bilateral adrenal hyperplasia (BAH) [1]. PA is the most common cause of secondary hypertension (HTN), affecting >5% and possibly >10% of hypertensive patients, in general and in specialty settings [1].

PA is important not only because of its prevalence, but also because of the higher risks of cardiovascular, renal, and cerebrovascular disease and metabolic syndrome in PA patients than in those with essential HTN [2,3]. This suggests that there are effects of aldosterone excess, independent of blood pressure (BP), on multiple organ systems [4]. Many studies conducted in animal and cell models have suggested that aldosterone excess can also impair glucose metabolism by impaired insulin secretion and by increased insulin resistance (IR) at several levels; hepatic glucose production and uptake, and glucose diffusion and uptake into insulin-sensitive tissues through a mineralocorticoid receptor (MR)-dependent and MR-independent mechanism [5,6,7,8,9,10,11,12,13,14]. Furthermore, the American Diabetes Association lists APA-induced hypokalemia as a secondary cause of type 2 diabetes mellitus (DM) [15]. However, clinical studies assessing the effects of PA on the prevalence of DM and glucose metabolism (e.g., elevated fasting plasma glucose (FPG), impaired insulin secretion, and increased IR), and the effects of the treatment modality used for PA, have generated conflicting results. First, the prevalence of DM was higher in PA patients than controls in several clinical studies [2,16,17], but not in others [18,19,20]. These inconsistent results may have been the result of differences in sample size, lack of sex, age, and body mass index (BMI)-matching, or no standardization for diagnosis of PA. Second, hypokalemia, a characteristic of PA, was not considered as a possible confounder in all these studies. Third, findings regarding parameters to assess glucose metabolism in PA were not fully consistent [16,21,22,23,24,25,26]. Fourth, recent studies showed that possible cortisol co-secretion rather than only aldosterone excess itself could contribute to the impairment of glucose metabolism in patients with PA [27,28]. Finally, comparisons of the effects of adrenalectomy (ADX) and mineralocorticoid receptor antagonist (MRA) therapy on glucose metabolism in PA have only sparsely been studied. The recent guideline recommends ADX for patients with documented unilateral PA (e.g., unilateral APA or UAH) or medical treatment including a MRA for patients are unable or unwilling to undergo surgery or for those with bilateral PA (e.g., bilateral APA or BAH) [1]. ADX can correct the aldosterone excess, whereas MRA can block the effects of aldosterone through MR-dependent mechanism and may not attenuate the effect of aldosterone excess. Previous clinical studies show discrepant results about comparisons of the effects of ADX and MRA therapy on glucose metabolism in PA [21,24,29,30]. Although one study showed no differences in decreased IR between the treatment modality (ADX vs. MRA therapy) [21], other clinical studies showed improved glucose metabolism in APA patients after ADX, but not in BAH patients after MRA therapy [24,29,30] and increased insulin secretion in APA patients after ADX [23,31]. However, previous clinical studies comparing the effects of ADX and MRA therapy used only small numbers of patients (*N* = 9–61), and therefore definitive conclusions cannot be drawn from these [11].

In an attempt to clarify these conflicting results, we compared glucose metabolism of 286 patients with PA that had been diagnosed using a standardized technique, and 816 controls, with a maximum 1:3 ratio of numbers, that had been matched with regard to sex, age (±1 year), and BMI (±0.5 kg/m^2^). We then compared the effect of two treatment modalities (ADX vs. MRA therapy) on the glucose status of PA patients.

## 2. Materials and Methods

### 2.1. Study Participants

Clinical data relating to PA patients were extracted using the Asan BiomedicaL research Environment, the de-identified clinical research data warehouse at the Asan Medical Center (AMC), Seoul, Korea [32], between May 2007 and April 2016. The study was approved by the Institutional Review Boards at AMC. The need for informed consent was waived because the clinical data of PA patients were extracted from Asan Biomedica Research Environment, which is a de-identified clinical data warehouse.

A plasma aldosterone concentration (PAC, ng/dL)/plasma renin activity (PRA, ng/mL/h) ratio (ARR) of ≥30 was used to screen PA cases [1]. Before measuring the PAC and PRA, all antihypertensive medications had been withdrawn for ≥4 weeks to prevent interference with the results. If absolutely necessary, subjects were administered an α-adrenergic blocker (e.g., doxazosin) and/or a non-dihydropyridine slow-release calcium channel blocker (e.g., verapamil), in accordance with the recent guidelines [1]. All patients had been encouraged to continue with oral potassium supplementation to help prevent hypokalemia, and there were no restrictions on the consumption of dietary salt before testing. The diagnosis of PA was confirmed by plasma aldosterone concentration (PAC) after intravenous saline infusion test (SIT: 2 L of 0.9% saline infused over 4 h) [1]. PAC after SIT > 10 ng/dL made the diagnosis of PA, whereas PAC after SIT < 5 ng/dL excluded the diagnosis PA. The SIT was repeated if PAC after SIT was indeterminate value between 5 and 10 ng/dL. However, PA was diagnosed without a confirmatory test in those patients with spontaneous hypokalemia, a PRA below the detection limit, or a PAC > 20 ng/dL [1]. Using these criteria, 286 PA patients were identified in the database (145 men and 141 women). To exclude PA patients with cortisol co-secretion, we used the criteria for diagnosing subclinical hypercortisolism (SH), subtle cortisol excess state, as cortisol level after a 1-mg overnight dexamethasone suppression test (1-mg DST) >5.0 μg/dL (138 nmol/L) or cortisol level after a 1-mg DST > 2.2 μg/dL (61 nmol/L) along with one parameter among low levels of adrenocorticotropic hormone (ACTH) (<10 pg/mL (2.2 pmol/L)) and dehydroepiandrosterone sulfate (DHEA-S) (<35 μg/dL (0.95 μmol/L) in women or <80 μg/dL (2.17 μmol/L) in men), as described previously [33]. Using the criteria for diagnosing SH, 245 of 286 (85.7%) PA patients were not SH status (e.g., cortisol co-secretion status). To form the control group for PA patients, 816 patients, with a maximum 1:3 ratio, who were matched on the basis of age, sex, and BMI, were selected from other patients visiting the Health Screening and Promotion Center at the AMC during the same period.

From the 286 PA cases, 81 PA patients who had not received any anti-diabetic therapies (28.3%, 42 men and 39 women), in whom serum insulin concentration was then measured, were selected from the Health Screening and Promotion Center, as a subgroup of patients in which parameters relating to glucose status were assessed. From the 816 matched controls, 225 who had not received any anti-diabetic therapies (27.5%, 114 men and 111 women) were also selected for the same analyses.

In accordance with the recent guideline [1], we recommended adrenal computed tomography in all 286 PA cases and adrenal venous sampling (AVS) when ADX was feasible and desired by the patient to make the distinction between unilateral PA (e.g., unilateral APA or UAH) and bilateral PA (e.g., bilateral APA or BAH). ADX was recommended for patients with unilateral PA, whereas MRA therapy was recommended for patients were unable or unwilling to undergo surgery or for patients with bilateral PA. In patients with unsuccessful AVS according to the selectivity index (SI) or those with intermediate values in lateralization index (LI) of between three and four in AVS findings [34], we used other clinical, biochemical, AVS findings (such as contralateral ratio and ipsilateral ratio), and ability or willingness to undergo surgery for reaching a therapeutic decision in an individual patient [1]. From the 286 PA patients, therefore, 178 PA patients who had undergone ADX were composed of 148 patients with unilateral APA, 20 patients with UAH, and 10 patients with unsuccessful AVS or those with intermediate values in LI. A total of 108 PA patients using MRA were composed of 66 patients with BAH, 10 patients with bilateral APA, six patients with unilateral APA, 14 patients with UAH, and 12 patients with unsuccessful AVS or those with intermediate values in LI. Information regarding frequency of outdoor exercise (< or ≥30 min/day), alcohol intake (< or ≥3 U/day), smoking habits (current smoker or not), previous medical or surgical procedures, history of use of medication, and menopausal status was obtained using an interview-assisted questionnaire.

### 2.2. Assessment of Glucose Metabolism

The presence of DM was recorded if the patient had a known history of DM, was being treated with an anti-diabetic therapy, and/or provided two blood samples with FPG ≥126 mg/dL (7.0 mmol/L) [35]. Pre-diabetes was recorded if the FPG was ≥100 mg/dL (5.6 mmol/L) and <126 mg/dL (7.0 mmol/L) in patients without DM [35]. The patient was recorded as being hyperglycemic if they had DM or pre-diabetes, according to these definitions. An improvement in glucose status was defined as a change in FPG from DM to pre-diabetic or normal glucose tolerance, or from pre-diabetic to normal glucose tolerance, or by a reduction in the anti-diabetic medications being administered.

The homeostasis model assessment (HOMA) was used to estimate β-cell function (HOMA-β) and insulin resistance (HOMA-IR) [36]. HOMA-IR was calculated using the following formula: (fasting insulin (μU/mL) × FPG (mg/dL))/405. HOMA-β was calculated using the following formula: (360 × fasting insulin (μU/mL))/(FPG (mg/dL) – 63).

### 2.3. Measurement of Serum Hormone Concentrations and Biochemical Parameters

Morning blood samples were drawn after an overnight fast. The PAC and PRA were measured by radioimmunoassay (SPAC-S aldosterone and PRA kits, respectively; TFB Inc., Tokyo, Japan), using a Cobra II Gamma Counter (Packard Instrument Co., Meriden, CT, USA). For the PAC assay, the lower limit of detection was 1.53 ng/dL, and the intra-assay and inter-assay coefficients of variation (CVs) were <3.2% and <6.7%, respectively. For the PRA assay, the lower limit of detection was 0.09 ng/mL/h, and the intra-assay and inter-assay CVs were <8.3% and <9.7%, respectively. The plasma ACTH levels were measured by immunoradiometric assay using an ELSA-ACTH kit (Cisbio bioassay; Codolet, Gard, France) on a Cobra II γ-counter (Packard Instrument Company, Meriden, CT, USA). Serum cortisol were measured by radioimmunoassay using the Coat-A-Count^®^ cortisol kit (Siemens Healthcare Diagnostics, Los Angeles, CA, USA) on a Cobra II γ-counter (Packard Instrument, Fallbrook, CA, USA). The DHEA-S level was measured by radioimmunoassay using the Coat-A-Count^®^ DHEA-SO4 kit (Siemens Healthcare Diagnostics, Los Angeles, CA, USA) on a Cobra II γ-counter (Packard Instrument, Fallbrook, CA, USA). We performed the 1-mg DST twice and used the mean value for analysis in those with a value of >1.8 μg/dL (50 nmol/L).

FPG was measured using the hexokinase method on an auto-analyzer (TBA-200FR; Toshiba, Tokyo, Japan). Serum insulin was measured using an immunoradiometric assay (TFB). Serum potassium concentration was measured using a Roche ISE Standard Low/High (Roche Diagnostics, Mannheim, Germany) ion selective electrode (ISE) and a Cobas 8000 ISE analyzer (Roche Diagnostics). The intra-assay and inter-assay CVs were 0.5% and 1.6%, respectively. Hypokalemia was defined by a serum potassium concentration <3.5 mmol/L. Serum creatinine was measured using a kinetic colorimetric assay, a Roche CREAJ2 kit (Roche Diagnostics), and the Cobas c702 module (Roche Diagnostics). The intra-assay and inter-assay CVs were <2.3% and <2.7%, respectively. Glomerular filtration rate was estimated using the Cockcroft–Gault Equation [37].

### 2.4. Statistical Analysis

Data are expressed as the mean ± standard deviation, the median (interquartile range), or number (percentage), unless stated otherwise. Baseline characteristics were compared using Student’s *t*-test or the Mann–Whitney *U-*test for continuous variables, and the χ^2^ test for categorical variables. Multiple logistic regression analyses were performed to calculate the odds ratio (OR) and 95% confidence intervals (95% CIs) for associations between the presence of PA and DM or hyperglycemia after including age, BMI, current smoking status, alcohol intake status, outdoor exercise status, and hypokalemia status as variables. Differences in FPG before and after ADX or MRA therapy were assessed using paired *t*-tests. The associations of PRA, PAC, ARR, or PAC after SIT with HOMA-β and HOMA-IR were investigated using multiple linear regression analyses. To generate the ORs (95% CIs) for improvements in glucose status according to treatment modality (ADX vs. MRA therapy), multiple logistic regression analyses were performed. For assessing whether associations between treatment modality (ADX vs. MRA therapy) (expressed as a categorical variable) and improvements in glucose metabolism were modified by sex (coded as 0 and 1 for women and men, respectively, and expressed as a categorical variable), interaction analysis was performed. All statistical analyses were performed using SPSS, version 22.0 (IBM, Inc., Armonk, NY, USA). A *p*-Value < 0.05 was deemed to represent statistical significance.

## 3. Results

Table 1 shows the baseline characteristics of the participants. PA patients (*N* = 286) had higher systolic BP (SBP), diastolic BP (DBP), and prevalence of HTN than their matched controls (*N* = 816), but lower serum potassium concentrations. DM was more prevalent among PA patients (19.6% vs. 13.1%, *p* = 0.011), as was pre-diabetes (36.0% vs. 24.5%, *p* < 0.001) (Table 1). DM was also more prevalent among PA patients without SH (*N* = 245) than among their matched control (*N* = 609; 18.8% vs. 12.9%, *p* = 0.024), as was pre-diabetes (36.7% vs. 24.7%, *p* = 0.002) (Appendix A). We also analyzed the data from the PA patients according to their hypokalemia status (Table 2). Of the PA patients, 25.5% (73/286) were hypokalemic, and those with hypokalemia were more likely to have DM (*p* = 0.008) or hyperglycemia (*p* = 0.001) than the matched controls. PA patients without hypokalemia were more likely to have pre-diabetes (*p* < 0.001) or hyperglycemia (*p* < 0.001) than the matched controls.

The 81 PA patients and the 73 PA patients without SH who had not been treated with anti-diabetic therapy had significantly lower serum insulin and HOMA-β than controls (all *p* < 0.001) (Appendix A). The PA patients and those without SH also had significantly lower HOMA-IR than controls (*p* < 0.001). However, PRA was not significantly associated with either HOMA-β (*p* = 0.817–0.942) or HOMA-IR (*p* = 0.248–0.447) (Appendix A). PAC tended to be inversely associated with HOMA-β (*p* = 0.058–0.093) and was inversely associated with HOMA-IR (*p* = 0.016–0.023). ARR was inversely associated with HOMA-IR (*p* = 0.044), but not with HOMA-β (*p* = 0.376), before adjustment. After adjustment for confounders, ARR was not significantly associated with either HOMA-IR (*p* = 0.092–0.104) or HOMA-β (*p* = 0.545–0.591). PAC after SIT was inversely associated with HOMA-β, before and after adjustment for confounders (*p* = 0.002–0.018), and tended to be inversely associated with HOMA-IR after adjustment for confounders (*p* = 0.063–0.070), but not before (*p* = 0.217).

PA patients were more likely to have DM (OR = 1.61, 95% CI = 1.13–2.30) than controls before adjustment (Model 1) in the Table 3. PA patients were also more likely to have DM (OR = 1.59–1.62, 95% CI = 1.10–2.38) than controls after adjustment for confounding factors (Model 2: sex, age, and BMI and Model 3: smoking status, alcohol intake, and regular outdoor exercise in addition to the variables included Model 2). Even after further adjustment for the presence of hypokalemia, which showed significant difference between PA patients and their matched controls, in addition to the variables included Model 3 (Model 4), PA patients were more likely to have DM (OR = 1.63, 95% CI = 1.11–2.39) than controls. PA patients were also more likely to be hyperglycemic (OR = 2.08, 95% CI = 1.58–2.73) than controls before adjustment (Model 1), and after adjustment for confounding factors, including hypokalemia (Models 2–4), the difference remained statistically significant (all *p* < 0.001). PA patients without SH were also more likely to have DM (OR = 1.56–1.65, 95% CI = 1.04–2.51) and hyperglycemia (OR = 2.06–2.11, 95% CI = 1.50–2.87) than controls before adjustment (Model 1) and after adjustment for confounding factors (Models 2–4).

We next classified the PA patients according to the treatment used (ADX and MRA therapy) and compared the effects of each (Appendix A). PA patients using MRA were more likely to take regular exercise than those who had undergone ADX (*p* = 0.009). PA patients who had undergone ADX had significantly lower serum potassium concentration (*p* = 0.003) and higher PAC, ARR, and PAC after SIT than those undergoing MRA therapy (all *p* < 0.001).

After ADX (*N* = 178), the prevalence of DM significantly decreased from 21.3% to 16.7% (*p* = 0.004) (Figure 1), but there was no significant change in the prevalence of DM in PA patients after MRA therapy (*N* = 108, *p* = 0.571). After ADX, FPG was also lower in PA patients (*p* = 0.006) and in PA patients who had not taken any anti-diabetic therapy (*p* < 0.001). After MRA therapy, FPG tended to be lower in PA patients (*p* = 0.093) and in PA patients who had not taken any anti-diabetic therapy (*p* = 0.053). After ADX in PA patients without SH (*N* = 148), the prevalence of DM significantly decreased from 19.6% to 8.1% (*p* = 0.004), but there was no significant change in the prevalence of DM in PA patients without SH after MRA therapy (*N* = 97, *p* = 0.558) (Appendix A). After ADX, FPG was also lower in PA patients without SH (*p* = 0.013) and in those PA patients without SH who had not taken any anti-diabetic therapy (*p* = 0.001). After MRA therapy, FPG did not change in PA patients without SH (*p* = 0.519) and in PA patients without SH who had not taken any anti-diabetic therapy (*p* = 0.352).

To clarify the difference by the subtype (unilateral PA vs. bilateral PA), we analyzed the effect of the treatment modality (ADX vs. MRA therapy) on glucose metabolism in 168 unilateral PA patients (148 patients with unilateral APA and 20 patients with UAH) of ADX group and in 76 bilateral PA patients (66 patients with BAH, 10 patients with bilateral APA) of MRA therapy group (Figure 2). The prevalence of DM in 168 unilateral PA patients after ADX significantly decreased from 20.8% to 10.7% (*p* = 0.011), but there was no significant change in the prevalence of DM in 76 bilateral PA patients after MRA therapy (*p* = 0.800). After ADX, FPG was also lower in unilateral PA patients (*p* = 0.002) and in unilateral PA patients who had not taken any anti-diabetic therapy (*p* < 0.001). After MRA therapy, FPG did not change in bilateral PA patients (*p* = 0.590) and in bilateral PA patients who had not taken any anti-diabetic therapy (*p* = 0.411).

PA patients who had undergone ADX had more improved glucose status (which defined as the change in glucose status from DM to pre-diabetes or normal glucose tolerance, or from pre-diabetes to normal glucose tolerance, or by the reduction of anti-diabetic medication) than those undergoing MRA therapy (OR = 1.95, 95% CI = 1.19–3.19) before adjustment (Model 1) in Table 4. After adjustment for confounding factors (Model 2 and Model 3), PA patients who had undergone ADX (OR = 1.91–2.02, 95% CI = 1.16–3.36) also had more improved glucose status than those undergoing MRA therapy.

Even after further adjustment for PAC after SIT, which showed significant difference between patients undergone ADX and those undergone MRA therapy, in addition to the variables included Model 3 (Model 5), PA patients who had undergone ADX (OR = 2.07, 95% CI = 1.10–3.90) had more improved glucose status than those undergoing MRA therapy. We then examined whether any relationship of treatment modality (ADX vs. MRA therapy) with improvements in glucose metabolism was modified by sex. We observed no interaction between treatment modality (ADX vs. MRA therapy) and sex for improvements in glucose metabolism (*p* for tests of interaction = 0.665–0.734). Therefore, there was no sex difference in associations between treatment modality and improvements in glucose metabolism. Furthermore, PA patients who had undergone ADX also had more improved glucose status than those undergoing MRA therapy, both before (Model 1) and after adjustment for confounders (Models 2, 3, and 5), within a similar range of PAC after SIT values (OR = 2.09–2.23, 95% CI = 1.14–4.27) (Appendix A). Unilateral PA patients who had undergone ADX had more improved glucose status than bilateral PA patients undergoing MRA therapy, both before (Model 1) and after adjustment for confounders (Models 2, 3, and 5) (OR = 1.81–1.99, 95% CI = 1.01–4.06) (Appendix A).

## 4. Discussion

The present study has shown that PA patients regardless of SH and hypokalemia status are at a higher risk of DM and pre-diabetes than sex-, age-, and BMI-matched controls. In the subgroup analysis, PA patients were shown to have lower insulin secretion than controls. After ADX in PA patients, the prevalence of DM and FPG was significantly lower, and ADX improved glucose status more effectively than MRA therapy. Furthermore, ADX for unilateral PA resulting in normalization of aldosterone excess also improved glucose metabolism more effectively than MRA therapy for bilateral PA resulting in remained aldosterone excess. Taking these findings together, we can conclude that glucose metabolism is impaired in PA independently of hypokalemia and subtle cortisol excess as a result of a reduction in insulin secretion by aldosterone excess through MR-independent mechanism, and that this problem was ameliorated more effectively by ADX than MRA therapy. To the best of our knowledge, this is the largest study to show superior effects of ADX over MRA therapy on glucose status in PA patients.

Previous studies have generated inconsistent results regarding the prevalence of DM in PA [2,16,17,18,19,20]. In the present study, the prevalence of DM (19.6%) and pre-diabetes (36.0%) in 286 PA patients was significantly higher than in the 815 matched controls enrolled from the Health Promotion Center in AMC (13.1% for DM and 24.5% for pre-diabetes) and in the general population, on the basis of data from the Korea National Health and Nutrition Examination Survey (13.0% for DM and 25.3% for pre-diabetes) [38]. Thus, PA patients have a higher risk of poor glucose status than the general population.

Hypokalemia was associated with lower insulin secretion from pancreatic β-cells in an in vitro study; therefore, the presence of hypokalemia in PA might confound investigations of the link between PA and glucose status [11,16,39]. Therefore, we also separately analyzed PA patients according to hypokalemia status and adjusted for hypokalemia. We found that hyperglycemia was more prevalent in the PA patients, regardless of hypokalemia status, than in the controls. Furthermore, PA patients had higher risks of hyperglycemia and DM than matched controls, even after adjustment for the presence of hypokalemia. These findings imply that the effect of aldosterone on glucose metabolism is independent of potassium status.

Only few data from human studies are available regarding the molecular mechanisms underlying the impairment in glucose metabolism in PA [26]. However, many studies conducted in animal and cell models suggest that aldosterone impairs glucose metabolism at several levels, including insulin secretion, hepatic glucose production and uptake, and glucose diffusion and uptake into insulin-sensitive tissues [5,6,7,8,9,10,11,12,13,14]. In the present study, the subgroup analysis (comprising ~28% of the study subjects) showed that HOMA-β was lower in PA patients than in matched controls. These findings are in agreement with data from clinical studies showing that PA patients have lower first phase insulin response, HOMA-β, and C-peptide concentration [23,25] and insulin secretion increased after ADX in PA patients [31], and those from in vitro and animal studies showing that aldosterone reduces insulin secretion. However, several clinical studies conducted in Caucasians have shown greater IR in PA [21,22,26]. Although the explanation for the differences in the results of parameters of glucose metabolism of patients with PA between the present study conducted in Korea and the previous studies conducted in Caucasians remains unclear, a higher risk of DM in lean individuals or a more substantial impairment in pancreatic β-cell function in Koreans than Caucasians [40,41,42] might be responsible. Consistent with this, PAC after SIT was inversely associated with HOMA-β in Korean PA patients. Taken together, these findings suggest that impairment in insulin secretion resulting from aldosterone excess might represent a mechanism for the poor glucose status in PA.

One further mechanism underlying the impairment in glucose metabolism in PA is glucocorticoid co-secretion. Although recent studies showed that higher prevalence of DM in PA patients might be associated mainly with SH [27,28], our result showed PA patients without SH, and thus with aldosterone excess alone, had a higher prevalence of DM than do sex-, age-, and BMI-matched controls. Interestingly, the DM prevalence and FPG levels were significantly decreased after ADX in PA patients without SH. This is in contrast to the recent study [28], which showed parameters of glucose metabolism did not improve at follow-up. However, this study included only 76 PA patients (25 patients after ADX and 48 patients after MRA therapy). ADX resulting in removal of only aldosterone excess in the our 148 PA patients without SH had a greater effect on glucose status than MRA therapy in our 97 PA patients without SH, so PA itself without subtle cortisol excess seemed to be associated with impaired glucose metabolism.

Data from studies in animal and cell models, aldosterone impairs insulin secretion and increases IR through MR-dependent and MR-independent mechanism [5,6,7,8,9,10,11,12,13,14], but regarding comparisons of the effects of ADX and MRA therapy on glucose metabolism in PA have only sparsely been studied [21,24,29,30]. One study showing no difference in decreased IR between ADX and MRA therapy [21] was in agreement with data from in vitro and animal studies showing MRA reduced IR via inhibited expression of pro-inflammatory factors and promoted expression of insulin-sensitizing factors such as peroxisome proliferator-activated receptor-γ and adiponectin in adipocytes [7] and via reduced inflammatory cytokines in the liver [13]. However, aldosterone impairs insulin secretion through MR-independent mechanism such as glucocorticoid receptor [6] or reactive oxygen species [8,10]. Therefore, MRA therapy can neither correct aldosterone excess nor improve impaired insulin secretion due to MR-independent mechanism of aldosterone. We speculate more improved glucose metabolism by ADX which can both correct aldosterone excess and improve impaired insulin secretion than by MRA therapy. Previous studies have suggested more beneficial effects of ADX than those of MRA therapy on glucose metabolism in PA [24,29,30] and increased insulin secretion after ADX [23,31] might support our speculation. However, only a few studies of small numbers of patients (*N* = 9–61) have been conducted to date. To our knowledge, we now present a greater effect on change in glucose metabolism by ADX than by MRA therapy in the PA patients with the largest number (*N* = 286). ADX significantly reduced the prevalence of DM and FPG in the PA patients with the largest numbers, but MRA therapy had no significant effects on these parameters. Furthermore, ADX improved glucose metabolism more than MRA therapy. To investigate whether the higher PAC after SIT in PA patients affects the differential effects of treatment modality, multiple logistic regression analyses were performed after additional adjustment for PAC after SIT, and in PA patients with a similar range of PAC after SIT values. In both analyses, ADX had a greater effect on glucose status than MRA therapy. We cannot determine the exact reason for this finding; normalization of aldosterone excess by ADX, but not by MRA therapy and MR-independent mechanism for impaired insulin secretion by aldosterone excess might be subsidiary reasons. Likewise, ADX for unilateral PA resulting in normalization of aldosterone excess also had more beneficial effects on glucose metabolism than MRA therapy for bilateral PA resulting in remained aldosterone excess. Selection of treatment modality (ADX vs. MRA therapy) was predisposed by the difference in PA subtype (unilateral PA vs. bilateral PA), so we supposed that PA subtypes might in part explain the difference in effects of treatment modality on the glucose metabolism. Thus, these findings indicate that ADX is the preferred treatment modality for its effects to ameliorate impairments in glucose metabolism in PA, which it achieves through a MR-independent mechanism.

The major strength of this study was that we analyzed data from a relatively large number of PA patients. This allowed us to perform a case-control study comparing patients and controls matched according to sex, age, and BMI, to standardize the diagnosis of PA, and to assess glucose status after treatment. Several potential limitations should also be considered when interpreting our data. First, we did not measure some parameters indicative of glucose status, for example, glycated hemoglobin, oral glucose tolerance test, or hyperinsulinemic–euglycemic clamp, in the participants, because these are not part of standard care for individuals who do not have DM. Second, we did not calculate HOMA-IR and HOMA-β values both at baseline and following treatment in the participants. Third, the data were obtained from patients at a single medical center in Korea, meaning that the findings are not readily generalizable to populations of differing ethnicity. Finally, selection of particular treatment modality was predisposed by the difference in PA subtypes, patient’s ability or willingness to undergo surgery, or other factors, so not all the PA patients undergoing MRA therapy had bilateral PA.

## 5. Conclusions

In summary, glucose status is impaired in PA patients, regardless of the hypokalemia and of the cortisol co-secretion, and is improved by ADX but not MRA therapy.

## Figures and Tables

**Figure 1 jcm-08-02194-f001:**
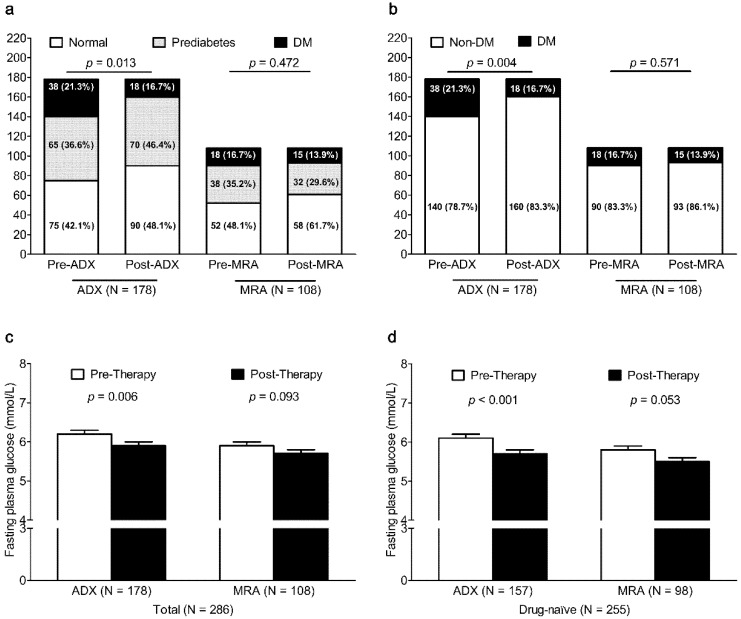
Change in glucose status and FPG in patients with PA after ADX or MRA therapy. Change in glucose status (normal, prediabetes, and DM in Figure (**a**); non-DM and DM in Figure (**b**) in patients with PA after ADX or MRA therapy. Change in FPG in patients with PA (Figure (**c**)) and that in anti-diabetic therapy naïve patients with PA (Figure (**d**)) after ADX or MRA therapy. ADX, adrenalectomy; DM, diabetes mellitus; FPG, fasting plasma glucose; MRA, mineralocorticoid receptor antagonist.

**Figure 2 jcm-08-02194-f002:**
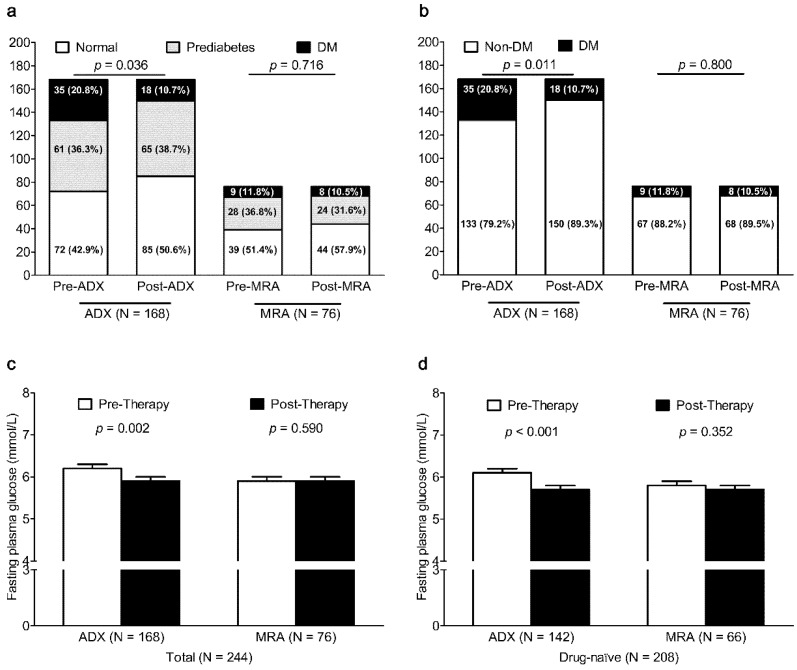
Change in glucose status and FPG in unilateral PA patients (unilateral APA and UAH) after ADX or that in bilateral PA patients (BAH and bilateral APA) after MRA therapy. Change in glucose status (normal, prediabetes, and DM in Figure (**a**); non-DM and DM in Figure (**b**) in unilateral PA patients after ADX and that in bilateral PA patients after MRA therapy. Change in FPG in unilateral PA patients (Figure (**c**)) and that in unilateral and anti-diabetic therapy naïve PA patients (Figure (**d**)) after ADX. Change in FPG in in bilateral PA patients (Figure c) and that in bilateral and anti-diabetic therapy naïve PA patients (Figure d) after MRA therapy. ADX, adrenalectomy; APA, aldosterone-producing adenoma; BAH, bilateral adrenal hyperplasia DM, diabetes mellitus; FPG, fasting plasma glucose; MRA, mineralocorticoid receptor antagonist; UAH, unilateral adrenal hyperplasia.

**Table 1 jcm-08-02194-t001:** Baseline characteristics of patients with PA and controls, matched at a ratio of up to 1:3.

	Control (*N* = 816)	PA (*N* = 286)	*p*
Men, *N* (%)	410 (50.3%)	145 (50.7%)	0.949
Age (years), mean ± SD	53.5 ± 11.1	53.3 ± 11.0	0.867
BMI (kg/m^2^), mean ± SD	25.2 ± 3.1	25.6 ± 3.7	0.109
Current smoker, *N* (%)	167 (20.5%)	52 (18.2%)	0.455
Alcohol intake ≥3 U/day, *N* (%)	119 (14.6%)	49 (17.1%)	0.349
Regular exercise ≥30 min/day, *N* (%)	359 (44.0%)	51 (17.8%)	<0.001
SBP (mmHg), mean ± SD	123.9 ± 16.8	139.9 ± 18.5	<0.001
DBP (mmHg), mean ± SD	80.2 ± 10.7	86.2 ± 12.0	<0.001
Hypertension, *N* (%)	309 (37.9%)	286 (100.0%)	<0.001
Potassium (mEq/L), mean ± SD	4.1 ± 0.4	3.9 ± 1.6	0.001
Creatinine (mg/dL), mean ± SD	0.8 ± 0.2	0.8 ± 0.2	0.416
eGFR (mL/min), mean ± SD	89.7 ± 22.6	97.2 ± 30.2	<0.001
FPG (mg/dL), mean ± SD	101.8 ± 25.3	110.9 ± 29.5	<0.001
FPG (mmol/L), mean ± SD	5.6 ± 1.4	6.2 ± 1.6	<0.001
Pre-diabetes, *N* (%)	200 (24.5%)	103 (36.0%)	<0.001
DM, *N* (%)	107 (13.1%)	56 (19.6%)	0.011
Hyperglycemia, *N* (%)	307 (37.6%)	159 (55.6%)	<0.001
Patients undergoing anti-diabetic therapy, *N* (%)	32 (4.5%)	29 (10.1%)	0.001
PRA (ng/mL/h), mean ± SD	NA	0.4 ± 0.9	
PAC (ng/dL), mean ± SD	NA	33.2 ± 11.8	
ARR, (ng/dL)/(ng/mL/h), mean ± SD	NA	171.4 ± 155.8	

Up to three controls were enrolled per participant with PA. They were individually matched for sex, age (±1 year), and BMI (±0.5 kg/m^2^). The hyperglycemia was defined as DM or pre-diabetes. Significant results (*p* < 0.05) are indicated in bold. ARR, aldosterone/renin ratio; BMI, body mass index; BP, blood pressure; DBP, diastolic blood pressure; DM, diabetes mellitus; GFR, glomerular filtration rat; FPG, fasting plasma glucose; NA, not applicable; PA, primary aldosteronism; PAC, plasma aldosterone concentration; PRA, plasma renin activity; SBP, systolic blood pressure.

**Table 2 jcm-08-02194-t002:** Glucose status in patients with PA and matched controls, classified according to the presence or absence of hypokalemia.

	Control (*N* = 212)	PA with hypoK (*N* = 73)	*p*	Control (*N* = 604)	PA without hypoK (*N* = 213)	*p*
Pre-diabetes, *N* (%)	49 (23.1%)	23 (31.5%)	0.155	151 (25.0%)	80 (37.6%)	<0.001
DM, *N* (%)	27 (12.7%)	19 (26.0%)	0.008	80 (13.2%)	37 (17.4%)	0.139
Hyperglycemia, *N* (%)	76 (35.8%)	42 (57.5%)	0.001	231 (38.2%)	117 (54.9%)	<0.001

Up to three controls were enrolled per participant with PA. They were individually matched for sex, age (±1 year), and BMI (±0.5 kg/m^2^). Hypokalemia was defined by a serum potassium concentration < 3.5 mEq/L. The hyperglycemia was defined as DM or pre-diabetes. Significant results (*p* < 0.05) are indicated in bold. BMI, body mass index; DM, diabetes mellitus; FPG, fasting plasma glucose; hypoK, hypokalemia; PA, primary aldosteronism.

**Table 3 jcm-08-02194-t003:** Risk of DM or hyperglycemia in the presence of PA.

	Control (*N* = 816), PA (*N* = 286)		Control (*N* = 699), PA without SH (*N* = 245)	
	OR (95% CI)	*p*	OR (95% CI)	*p*
DM				
Model 1	1.61 (1.13–2.30)	0.008	1.56 (1.06–2.31)	0.024
Model 2	1.59 (1.10–2.30)	0.013	1.56 (1.04–2.33)	0.031
Model 3	1.62 (1.10–2.38)	0.014	1.64 (1.08–2.50)	0.020
Model 4	1.63 (1.11–2.39)	0.013	1.65 (1.08–2.51)	0.020
Hyperglycemia				
Model 1	2.08 (1.58–2.73)	<0.001	2.07 (1.54–2.78)	<0.001
Model 2	2.12 (1.59–2.82)	<0.001	2.11 (1.55–2.87)	<0.001
Model 3	2.08 (1.55–2.79)	<0.001	2.06 (1.50–2.83)	<0.001
Model 4	2.08 (1.55–2.79)	<0.001	2.06 (1.50–2.83)	< 0.001

The hyperglycemia was defined as DM or pre-diabetes. Model 1: unadjusted model. Model 2: adjusted for sex, age, and BMI. Model 3: adjusted for current smoking status, alcohol intake (≥3 units/day), and regular outdoor exercise (≥30 min/day), in addition to the variables included in Model 2. Model 4: adjusted for the presence of hypokalemia (K < 3.5 mEq/L), in addition to the variables included in Model 3.

**Table 4 jcm-08-02194-t004:** Comparison of the improvement in glucose status achieved by treatment with ADX vs. MRA therapy.

	OR (95% CI)	*p*
Model 1		
MRA therapy	Ref.	
ADX	1.95 (1.19–3.19)	0.008
Model 2		
MRA therapy	Ref.	
ADX	1.91 (1.16–3.14)	0.011
Model 3		
MRA therapy	Ref.	
ADX	2.02 (1.21–3.36)	0.007
Model 5		
MRA therapy	Ref.	
ADX	2.07 (1.10–3.90)	0.024

Significant results (*p* < 0.05) are indicated in bold. Model 1: unadjusted model; Model 2: adjusted for sex, age, and BMI; Model 3: adjusted for current smoking status, alcohol intake (≥3 units/day), and regular outdoor exercise (≥30 min/day), in addition to the variables included in Models 2 and 5: adjusted for plasma aldosterone concentration after intravenous saline infusion test, in addition to the variables included in Model 3. Improvement in glucose status is defined as the change in glucose status from diabetes mellitus to pre-diabetes or normal glucose tolerance, or from pre-diabetes to normal glucose tolerance, or by the reduction of anti-diabetic medication. ADX, adrenalectomy; BMI, body mass index; MRA, mineralocorticoid receptor antagonist; OR, odds ratio; PA, primary aldosteronism; 95% CI, 95% confidence interval.

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
