# Peer review of "Effects of Primary Aldosteronism and Different Therapeutic Modalities on Glucose Metabolism"

_jcm, 2019, doi:10.3390/jcm8122194_

Round 1

Reviewer 1 Report

I have reviewed current manuscript (jcm-638763) as the Reviewer 1. In the current revised manuscript (jcm-665433), the authors addressed my critiques, comments and suggestions.  Hence, the revised manuscript (jcm-665433) could be accepted as it is.

Reviewer 2 Report

I have read the revised version. The authors have addressed my comments and I accept the article for publication.

This manuscript is a resubmission of an earlier submission. The following is a list of the peer review reports and author responses from that submission.

Round 1

Reviewer 1 Report

The major strength of this well designed retrospective study was data analyzed from a large number of patients with PA in a single center, which allowed authors to compare patients and controls matched according to sex, age, and BMI, to standardize the diagnosis of PA, and to assess glucose status after treatment.

The major concern of this retrospective study: it is ulcer how patients with PA were selected for particular therapy - ADX vs. MRA. Other words, what were the difference in enrolment criteria for treatments. Also unclear if the particular treatment selection was predisposed by the difference in PA causes such as aldosterone-producing adenoma (APA) vs. unilateral or bilateral adrenal hyperplasia (BAH) or other. This knowledge’s are critical, since they might in part or totally explain the difference in treatments (ADX vs. MRA) outcomes with respect to glucose metabolism and insulin secretion. Hence, this information should be incorporated into the manuscript or described as one of the major limitations.

Abstract:

Better to say:

-          Lane 14: “Despite findings” instead of “Despite studies”.

-          Lane 15: “controversial” instead of “inconsistent”.

-          Line 18: Treatment should be specified:  adrenalectomy and mineralocorticoid receptor antagonist (MRA) therapy (spironolactone).

Lowercase italic “p” should be used instead of “P”.

Introduction:

Existing treatment modality (surgical removal of the gland vs. aldosterone clocking drugs?) used for PA should be described; and mechanism of action of aldosterone through mineralocorticoid receptors should be clearly introduced, otherwise is not completely unclear why authors decided to compare outcomes of adrenalectomy (ADX) and mineralocorticoid receptor antagonist (MRA) therapy on glucose metabolism in PA.

Reference # 31 should be addressed in Introduction and not only at the end of Discussion.

Another important publication on presenting topic - Sindelka G, Widimsky J, Haas T, Prazny M, Hilgertova J, Skrha J. Insulin action in primary hyperaldosteronism before and after surgical or pharmacological treatment. Exp Clin Endocrinol Diabetes 108: 21–25, 2000 – is not included into Introduction/Discussion.

Results:

Lane 62-63: Treatment modalities should be specified in the following sentence: “We then compared the effect of two treatment modalities on the glucose status of PA patients”

PAC and PRA data should be included into Table 1 that presentsBaseline characteristics of patients with PA and controls”.

If any sex difference in treatment outcomes with respect to glucose metabolism were detected?

Discussion:

It is known that aldosterone blockade in general improves glucose homeostasis, however this topic is not properly addressed in the Discussion.

List of abbreviations is required.

Reviewer 2 Report

The authors have conducted a large patient study with PA and determined correlation of PA with overt Diabetes Mellitus or pre-diabetes. The strength of the study compared to previous studies lies in the large number of patients which has given significant power to their analyses. The study is interesting in that they found ADX improved glucose metabolism in the patients.

Inclusion of couple of sentences in the discussion section on how aldosterone inhibits insulin secretion in cell lines and animal models besides just citing papers and juxtaposing it to how it may act in humans (based on the results you obtained) would add to the content of the paper.